# DECENTRALIZED KNOWLEDGE GRAPH REPRESENTATION LEARNING

## ABSTRACT

Knowledge graph (KG) representation learning methods have achieved competitive performance in many KG-oriented tasks, among which the best ones are usually based on graph neural networks (GNNs), a powerful family of networks that learns the representation of an entity by aggregating the features of its neighbors and itself. However, many KG representation learning scenarios only provide the structure information that describes the relationships among entities, causing that entities have no input features. In this case, existing aggregation mechanisms are incapable of inducing embeddings of unseen entities as these entities have no pre-defined features for aggregation. In this paper, we present a decentralized KG representation learning approach, decentRL, which encodes each entity from and only from the embeddings of its neighbors. For optimization, we design an algorithm to distill knowledge from the model itself such that the output embeddings can continuously gain knowledge from the corresponding original embeddings. Extensive experiments show that the proposed approach performed better than many cutting-edge models on the entity alignment task, and achieved competitive performance on the entity prediction task. Furthermore, under the inductive setting, it significantly outperformed all baselines on both tasks.

## 1 INTRODUCTION

Knowledge graphs (KGs) support many data-driven applications (Ji et al., 2020). Recently, learning low-dimensional representations (a.k.a. embeddings) of entities and relations in KGs has been increasingly given attentions (Rossi et al., 2020). We find that existing models for KG representation learning share similar characteristics to those for word representation learning. For example, TransE (Bordes et al., 2013), a well-known translational KG embedding model, interprets a triple $(e_1, r, e_2)$ as $\mathbf{e}_1 + \mathbf{r} \approx \mathbf{e}_2$, where $e_1, e_2, r$ denote subject, object and their relationship, respectively, and the boldfaces denote the corresponding embeddings. If we view $e_1$ as a word in sentences, and $e_2$ as well as many other objects of $e_1$ as the context words, then TransE and many KG embedding models (Wang et al., 2014; Dettmers et al., 2018; Nguyen et al., 2018; Kazemi & Poole, 2018; Sun et al., 2019), learn representations in a form similar to that used in Skip-gram (Mikolov et al., 2013a), where the input representation is learned to predict the context (i.e., neighbors) representations.

Recently, many graph neural networks (GNNs) based models for KG representation learning (Wang et al., 2018; Schlichtkrull et al., 2018; Cao et al., 2019; Wu et al., 2019; Sun et al., 2020; Vashishth et al., 2020) have achieved state-of-the-art performance in KG-related tasks such as entity alignment and entity prediction. Those models learn KG representations in a CBOW (continuous bag-of-words) (Mikolov et al., 2013a) manner, in which the context entities are aggregated to predict the target. But they also consider the representation of an entity itself when aggregating the neighborhood information. This nature prevents those models (e.g., GCN (Kipf & Welling, 2017) and GAT (Velickovic et al., 2018)) to be generalized to represent unseen entities. In many cases, the entities in prevalent KG-related tasks do not have self features. This motivates us to learn entity representations from and only from their context neighbors.

We propose a decentralized KG representation learning approach, **decentRL**. The key idea of decentRL is to decentralize the semantic information of entities over only their neighbors (i.e., distributed context vector in CBOW (Mikolov et al., 2013b)), which can be easily implemented by representing each entity through averaging its neighbor embeddings. In this paper, we look for

a more efficient but still simple way to realize this concept on the most popular graph attention network (GAT) (Velickovic et al., 2018), as well as its many variants (Sun et al., 2020; Vashishth et al., 2020). We illustrate the methodology by decentralized attention network (DAN), which is based on the vallina GAT. DAN is able to support KG representation learning for unseen entities with only structure information, which is essentially different from the way of using self features (e.g., attribute information) in the existing graph embedding models (Hamilton et al., 2017; Bojchevski & Günnemann, 2018; Hettige et al., 2020). Furthermore, the neighbors in DAN serve as an integrity to give attentions, which means that DAN is more robust and more expressive compared with conventional graph attention mechanism (Velickovic et al., 2018).

Another key problem in decentralized KG representation learning is how to estimate and optimize the output embeddings. If we distribute the information of an entity over its neighbors, the original embedding of this entity $\mathbf{e}_i$ also learns how to effectively participate in the aggregations of its different neighbors conversely. Suppose that we have obtained an output representation $\mathbf{g}_i$ from DAN for entity $e_i$, we can simply estimate and optimize $\mathbf{g}_i$ by aligning it with $\mathbf{e}_i$. But directly minimizing the L1/L2 distance between $\mathbf{g}_i$ and $\mathbf{e}_i$ may be insufficient. Specifically, these two embeddings have completely different roles and functions in the model, and the shared information may not reside in the same dimensions. Therefore, maximizing the mutual information between them is a better choice. Different from the existing works like MINE (Belghazi et al., 2018) or InfoNCE (van den Oord et al., 2018), in this paper, we design a self knowledge distillation algorithm, called auto-distiller. It alternately optimizes $\mathbf{g}_i$ and its potential target $\mathbf{e}_i$, such that $\mathbf{g}_i$ can automatically and continuously distill knowledge from the original representation $\mathbf{e}_i$ across different batches.

The main contributions of this paper are listed as follows. (1) We propose decentralized KG representation learning, and present DAN as the prototype of graph attention mechanism under the open-world setting. (2) We design an efficient knowledge distillation algorithm to support DAN for generating representations of unseen entities. (3) We implement an end-to-end framework based on DAN and auto-distiller. The experiments show that it achieved superior performance on two prevalent KG representation learning tasks (i.e., entity alignment and entity prediction), and also significantly outperformed those cutting-edge models under the open-world setting.

## 2    BACKGROUND

**Knowledge Graph.** A KG can be viewed as a multi-relational graph, in which nodes represent entities in the real world and edges have specific labels to represent different relationships between entities. Formally, we define a KG as a 3-tuple $\mathcal{G} = (\mathcal{E}, \mathcal{R}, \mathcal{T})$, with $\mathcal{E}$ and $\mathcal{R}$ denoting the sets of entities and relationships, respectively. $\mathcal{T}$ is the set of relational triples.

**KG Representation Learning**. Conventional models are mainly based on the idea of Skip-gram. According to the types of their score functions, these models can be divided into three categories: translational models (e.g., TransE (Bordes et al., 2013) and TransR (Lin et al., 2015a)), semantic matching models (e.g., DistMult (Yang et al., 2015) and ComplEx (Trouillon et al., 2016)) and neural models (e.g., ConvE (Dettmers et al., 2018) and RSN (Guo et al., 2019)). We refer interested readers to the surveys (Wang et al., 2017; Ji et al., 2020) for details. Recently, GNN-based models receive great attentions in this field, which are closely related to this paper. Specifically, R-GCN (Schlichtkrull et al., 2018), AVR-GCN (Ye et al., 2019) and CompGCN (Vashishth et al., 2020) introduce different relation-specific composition operations to combine neighbors and the corresponding relations before neighbor aggregation. RDGCN (Wu et al., 2019) refactors KGs as dual relation graphs (Monti et al., 2018) where edge labels are represented as nodes for graph convolution. All the aforementioned GNN-based models choose GCNs and/or GATs to aggregate the neighbors of an entity, in which an identity matrix is added to the adjacency matrix. This operation is helpful when elements have self features, but poses a problem in learning the representations of unseen entities where no self features are attached to them. Differently, decentRL fully relies on the neighbor context to attend to the neighbors of each entity in linear complexity, which is efficient and easy to be deployed.

**Entity Alignment.** Entity alignment aims to find the potentially aligned entity pairs in two different KGs $\mathcal{G}_1 = (\mathcal{E}_1, \mathcal{R}_1, \mathcal{T}_1)$ and $\mathcal{G}_2 = (\mathcal{E}_2, \mathcal{R}_2, \mathcal{T}_2)$, given a limited number of aligned pairs as training data $\mathcal{S} \subset \mathcal{E}_1 \times \mathcal{E}_2$. Oftentimes, $\mathcal{G}_1, \mathcal{G}_2$ are merged to a joint KG $\mathcal{G} = (\mathcal{E}, \mathcal{R}, \mathcal{T})$, which enables the models learn representations in a unified space.

**Entity Prediction.** Entity prediction (a.k.a. KG completion (Bordes et al., 2013)) seeks to find the missing subject $e_1$ or object $e_2$, given an incomplete relation triple $(?, r, e_2)$ or $(e_1, r, ?)$.

It is worth noting that the performance on the entity prediction task may be greatly improved by complex deep networks, as it relies on the predictive ability rather than the embedding quality (Guo et al., 2019). Hence, many cutting-edge models cannot obtain promising results in entity alignment (Guo et al., 2019; Sun et al., 2020). Differently, entity alignment directly compares the distance of learned entity embeddings, which clearly reflects the quality of output representations. Few models demonstrate consistently good performance on both tasks, whereas decentRL is capable of achieving competitive, even better, performance compared with respective state-of-the-art models.

## 3 DECENTRALIZED REPRESENTATION LEARNING

In the decentralized setting, the representation of an entity $e_i$ is aggregated from and only from its neighbors $N_i = \{e_1, e_2, \ldots, e_{|N_i|}\}$. As it may have many neighbors that are unequally informative (Velickovic et al., 2018), involving attention mechanism is a good choice.

### 3.1 GRAPH ATTENTION NETWORKS

We start by introducing the Graph attention network (GAT) (Velickovic et al., 2018), which leverages linear self attention to operate spatially close neighbors. For an entity $e_i$, GAT aggregates the representations of its neighbors $N_i$ and itself into a single representation $\mathbf{c}_i$ as follows:

$$\mathbf{c}_i = \sum_{e_j \in N_i \cup \{e_i\}} a_{ij} \mathbf{W} \mathbf{e}_j, \tag{1}$$

where $a_{ij}$ is the learnable attention score from $e_i$ to $e_j$, and $\mathbf{W}$ is the weight matrix. To obtain $a_{ij}$, a linear attention mechanism is used here:

$$a_{ij} = softmax\big(\sigma(\mathbf{a}^{\mathsf{T}}[\mathbf{W}_1 \mathbf{e}_i \,\|\, \mathbf{W}_2 \mathbf{e}_j])\big), \tag{2}$$

where $\mathbf{a}$ is a weight vector to convert the concatenation of two embeddings into a scalar attention score, and $\|$ denotes the concatenation operation. $\mathbf{W}_1$ and $\mathbf{W}_2$ are two weight matrices. $\sigma$ is the activation function, usually being LeakyReLU (Xu et al., 2015). GAT computes the attention score of an entity $e_i$ to its neighbors in linear complexity, which is very efficient when being applied to large-scale graphs.

### 3.2 DECENTRALIZED ATTENTION NETWORKS

Intuitively, if $\mathbf{e}_i$ is the embedding of an unseen entity, it is rarely useful in computing the attention scores (as it is just a randomly initialized vector). Thus, purely relying on its neighbors may be a good choice. Specifically, to obtain the decentralized attention scores, one may simply sum all the attention scores from other neighbors $a'_{ij} = softmax(\sum_{e_k \in N_i \setminus \{e_j\}} a_{kj})$. However, it would lead to a problem that this sum only represents the attention of each neighbor to $e_j$. In this case, a high attention score from one neighbor $e_k$ to $e_j$ can dominate the value of $a'_{ij}$, but it does not mean that $e_j$ is more important for $e_i$. Therefore, all neighbors should act as an integrity in giving attentions.

Towards this end, we propose decentralized attention networks (DANs). Formally, to obtain the decentralized attention weight $a_{ij}$, we have to feed the attention layer with two types of input: the neighbor context vector $\mathbf{n}_i$ (i.e., query), and the candidate neighbor embedding $\mathbf{e}_j$ (i.e., key and value). Separately controlling the iterations of these two variables in a multi-layer model is evidently inefficient. Instead, we realize this operation by involving a second-order attention mechanism. For layer $k$, DAN calculates the decentralized attention score $a_{ij}^k$ as follows:

$$a_{ij}^k = softmax\big(\sigma(\mathbf{a}_k^{\mathsf{T}}[\mathbf{W}_1^k \mathbf{d}_i^{k-1} \,\|\, \mathbf{W}_2^k \mathbf{d}_j^{k-2}])\big), \tag{3}$$

where $\mathbf{d}_i^{k-1}, \mathbf{d}_j^{k-2}$ denote the output embeddings of layer $k-1$ for $e_i$ and of layer $k-2$ for $e_j$, respectively. If we regard $\mathbf{d}_i^{k-1}$ as the neighbor aggregation of layer $k-1$ for $e_i$, then $\mathbf{d}_j^{k-2}$ is exactly the embedding of $e_j$ used in summing $\mathbf{d}_i^{k-1}$. In this case, $a_{ij}^k$ can represent the attention weight of

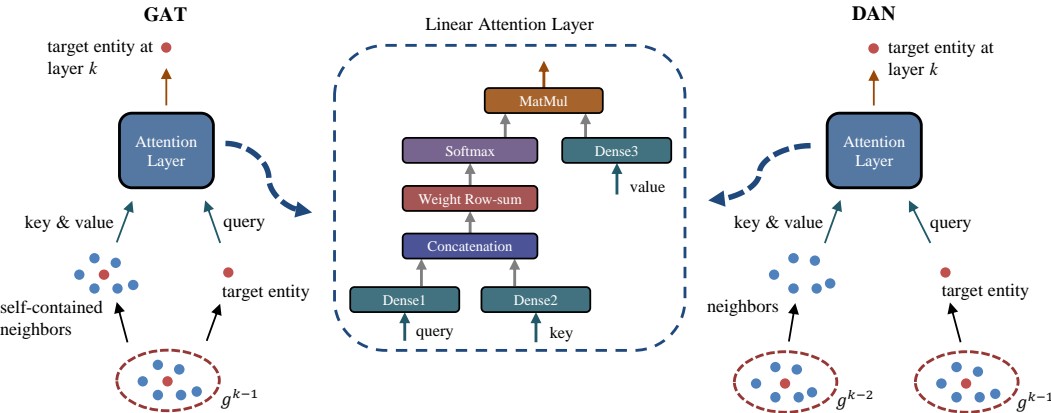

Figure 1: Comparing graph attention network (GAT) with decentralized attention network (DAN).

$e_i$'s neighbor context to $e_j$. Then, we can obtain the output of layer $k$ by:

$$\mathbf{d}_i^k = \sum_{e_j \in N_i} a_{ij}^k \mathbf{W}^k \mathbf{d}_j^{k-2}. \tag{4}$$

It is worth noting that we perform convolutions on layer $k - 2$, as the score $a_{ij}^k$ is attended to the neighbor representations in layer $k - 2$. This keeps the consistency and ensures the output representations are consecutive. Also, it enhances the correlation of output in different layers, and forms the second-order graph attention mechanism.

For the first layer of DAN, we initialize $\mathbf{d}_i^0$ and $\mathbf{d}_j^{-1}$ as follows:

$$\mathbf{d}_i^0 = \frac{1}{|N_i|} \sum_{e_j \in N_i} \mathbf{W}^0 \mathbf{e}_j, \quad \mathbf{d}_j^{-1} = \mathbf{e}_j. \tag{5}$$

Here, we simply use a mean aggregator to obtain the decentralized embedding $\mathbf{d}_i^0$ of layer 0, but other aggregators like pooling may be employed as well. This simple mean aggregator can also be regarded as a CBOW model with dynamic window size. For the architecture and implementation of DAN, please refer to Appendix A.

## 3.3 INSIGHT OF DAN

We compare GAT with DAN in Figure 1. Although the attention layers leveraged by DAN and GAT are identical, the decentralized structure has two significant strengths:

**Inductive representation learning.** In GAT, the self embedding $\mathbf{e}_i$ participates in the calculation of attention scores (Equation 1) and the aggregation of neighbors (Equation 2). Therefore, when $e_i$ is an open entity, its embedding is completely random-initialized. In this case, the attention scores computed by GAT are almost meaningless. By contrast, DAN generates the embedding of $e_i$ without requirement of its embedding throughout. Such characteristic enables DAN to induce embeddings on unseen entities.

**Robustness.** When calculating the attention scores for an entity $e_i$, GAT only takes $\mathbf{e}_i$ as query, the importance of other neighbors is overlooked, which may lead to biased attention computation. On the other hand, it is generally known that most entities in KGs only has a small number of neighbors (Li et al., 2017). Due to the lack of training examples, the embeddings of these entities are not as informative as those with more neighbors. Therefore, they may be not capable of serving as queries for computing attention scores, causing that GAT cannot obtain reliable attention scores in some cases. By contrast, the queries in DAN are neighbor context vectors, which have richer semantics and also enable DAN to compute the attention scores in an unbiased way.

Furthermore, the computational complexity of DAN is almost identical to that of GAT, except that DAN has an additional mean aggregator. From Figure 1 and Equation 5, we can find that such an aggregator is evidently simpler than the linear attention layer, which means that its computational complexity (both time and space) can be almost overlooked. Therefore, DAN is an efficient model.

## 4    DECENTRALIZED REPRESENTATION ESTIMATION

The final output representation $\mathbf{g}_i$ of DAN for $e_i$ can be optimized by minimizing the L1/L2 distance between $\mathbf{g}_i$ and $\mathbf{e}_i$ to enable self-supervised learning. Such distance estimation pursues a precise match at every dimension of these two embeddings, but ignores the implicit structure information across different dimensions.

### 4.1    MUTUAL INFORMATION MAXIMIZATION

As mentioned in Section 1, the original embedding $\mathbf{e}_i$ also serves as one of neighbor embeddings in aggregating the decentralized embeddings of its neighbors, which implies that $\mathbf{e}_i$ itself also preserves the latent information used to support its neighbors. Inspired by MINE (Belghazi et al., 2018), InfoNCE (van den Oord et al., 2018), and DIM (Hjelm et al., 2019), in this paper we do not try to optimize $\mathbf{g}_i$ by reconstructing the original representation $\mathbf{e}_i$. Instead, we implicitly align them in a way of maximizing the mutual information $I(\mathbf{g}_i, \mathbf{e}_i)$.

Specifically, we define a learnable function $f : \mathbb{R}^D \otimes \mathbb{R}^O \to \mathbb{R}$ to estimate the mutual information density (van den Oord et al., 2018) between $\mathbf{g}_i$ and the copy of $\mathbf{e}_i$ (the reason why using the copied vector will be explained shortly):

$$f(\mathbf{g}_i, \mathbf{e}_i) = \exp(\mathbf{g}_i^\mathsf{T} \mathbf{W}_f \hat{\mathbf{e}}_i + \mathbf{b}_f), \tag{6}$$

where $D$ and $O$ are the dimensions of the output and input representations, respectively. $\mathbf{W}_f, \mathbf{b}_f$ are the weight matrix and bias, respectively. $\hat{\mathbf{e}}_i$ denotes the copy of $\mathbf{e}_i$. We expect that $f(\mathbf{g}_i, \hat{\mathbf{e}}_i)$ is significantly larger than $f(\mathbf{g}_i, \hat{\mathbf{e}}_j)$ for $j \neq i$. Following InfoNCE, the objective can be written as:

$$\widehat{I}(\mathbf{g}_i, \hat{\mathbf{e}}_i) = \mathop{\mathbb{E}}_{X_i} \log \Big( \frac{f(\mathbf{g}_i, \hat{\mathbf{e}}_i)}{\sum_{e_j \in X_i} f(\mathbf{g}_i, \hat{\mathbf{e}}_j)} \Big), \tag{7}$$

where $X_i = \{e_1, \ldots, e_{|X_i|}\}$ contains $|X_i| - 1$ sampled negative entities plus the target entity $e_i$. Maximizing this objective results in maximizing a lower-bound of mutual information between $\mathbf{g}_i$ and $\hat{\mathbf{e}}_i$ (van den Oord et al., 2018).

### 4.2    AUTO-DISTILLER

Note that in Equations (6) and (7), we actually use the copy of the original representations, which leads to a completely different optimization process compared with existing works. Specifically, some methods like InfoNCE or DIM jointly optimize the two input in the density function, as both variables are the output of deep neural models requiring to be updated in back-propagation. But in decentRL, $\mathbf{e}_i$ is just a randomly initialized vector, and its gradient in Equation (7) may conflict with the gradient where it is taken as an input neighbor in learning decentralized representations of its neighbors. On the other hand, such optimization operation also prevents $\mathbf{e}_i$ from learning the neighborhood information at the bottom layer, and compels this variable to match $\mathbf{g}_i$.

To address this problem, we view $\mathbf{e}_i$ as a teacher, and $\mathbf{g}_i$ as a student to learn from the teacher (Tian et al., 2020). Our aim is to let this teacher continuously gain more knowledge to teach the student, which we call it auto-distiller. Therefore, our final objective is:

$$\underset{\mathbf{g}_i, f}{\mathrm{argmax}} \, \mathop{\mathbb{E}}_{X_i} \log \Big( \frac{f(\mathbf{g}_i, \hat{\mathbf{e}}_i)}{\sum_{e_j \in X_i} f(\mathbf{g}_i, \hat{\mathbf{e}}_j)} \Big) + \underset{\mathbf{e}_i}{\mathrm{argmax}} \sum_{e_j \in N_i} \mathop{\mathbb{E}}_{X_j} \log \Big( \frac{f(\mathbf{g}_j, \hat{\mathbf{e}}_j)}{\sum_{e_k \in X_j} f(\mathbf{g}_j, \hat{\mathbf{e}}_k)} \Big), \tag{8}$$

and it has two important characteristics:

**Lemma 1** (automatic distillation). *Optimizing the first term of Equation 8 for entity $e_i$ naturally contributes to optimizing the second term for the neighbors of $e_i$, which means conventional mini-batch training procedure can be applied.*

**Lemma 2** (lower-bound). *The mutual information between $\mathbf{g}_i$ and $\hat{\mathbf{e}}_i$ is still lower-bounded in auto-distiller.*

*Proof.* See Appendix B.    □

Table 1: Result comparison of entity alignment on DBP15K.

| Models | ZH-EN | | | JA-EN | | | FR-EN | | |
|---|---|---|---|---|---|---|---|---|---|
| | Hits@1 | Hits@10 | MRR | Hits@1 | Hits@10 | MRR | Hits@1 | Hits@10 | MRR |
| AlignE (Sun et al., 2018) | 0.472 | 0.792 | 0.581 | 0.448 | 0.789 | 0.563 | 0.481 | 0.824 | 0.599 |
| RSN (Guo et al., 2019) | 0.508 | 0.745 | 0.591 | 0.507 | 0.737 | 0.590 | 0.516 | 0.768 | 0.605 |
| GAT (Velickovic et al., 2018) | 0.418 | 0.667 | 0.508 | 0.446 | 0.695 | 0.537 | 0.442 | 0.731 | 0.546 |
| AliNet (Sun et al., 2020) | 0.539 | **0.826** | 0.628 | 0.549 | **0.831** | 0.645 | 0.552 | **0.852** | 0.657 |
| decentRL | **0.589** | 0.819 | **0.672** | **0.596** | 0.819 | **0.678** | **0.602** | 0.842 | **0.689** |

## 5 EXPERIMENTS

We evaluated decentRL on two prevalent tasks, namely entity alignment and entity prediction, for KG representation learning. As few existing models show state-of-the-art performance on both tasks, we picked the state-of-the-art methods in respective tasks and compared decentRL with them. To probe the effectiveness of decentRL, we also conducted ablation study and additional experiments. Limited by the length, please see Appendix C for more analytic results.

### 5.1 DATASETS

**Entity Alignment Datasets.** We consider the JAPE dataset DBP15K (Sun et al., 2017), which is widely used by existing studies. It includes three entity alignment settings, each of which contains two KGs of different languages. For example, ZH-EN indicates Chinese-English alignment on DBpedia.

**Entity Prediction Datasets.** We consider four datasets: FB15K, WN18, FB15K-237, and WN18RR (Bordes et al., 2013; Dettmers et al., 2018; Toutanova & Chen, 2015). The former two have been used as benchmarks for many years, while the latter two are the corrected versions, as FB15K and WN18 contain a considerable amount of redundant data (Dettmers et al., 2018).

### 5.2 EXPERIMENT SETUP

For both tasks, we initialized the original entity embeddings, relation embeddings and weight matrices with Xavier initializer (Glorot & Bengio, 2010).

To learn cross-KG embeddings for the entity alignment task, we incorporated a contrastive loss (Sun et al., 2020; Wang et al., 2018) to cope with aligned entity pairs $\mathcal{S}$, which can be written as follows:

$$\mathcal{L}_a = \sum_{(i,j)\in\mathcal{S}^+} ||\mathbf{g}_i - \mathbf{g}_j|| + \sum_{(i',j')\in\mathcal{S}^-} \alpha\big[\lambda - ||\mathbf{g}_{i'} - \mathbf{g}_{j'}||\big]_+, \tag{9}$$

where $\mathcal{S}^+$, $\mathcal{S}^-$ are the positive entity pair set and sampled negative entity pair set, respectively. $|| \cdot ||$ denotes the L2 distance between two embeddings. $\alpha$ and $\lambda$ are hyper-parameters. By jointly minimizing two types of losses, decentRL is able to learn cross-KG embeddings for entity alignment.

Similarly, for entity prediction, we also need to choose a decoder to enable decentRL to predict missing entities (Vashishth et al., 2020). We chose two simple models, TransE (Bordes et al., 2013) and DistMult (Yang et al., 2015) for the main experiments, which are sufficient to achieve comparable performance against the state-of-the-art.

### 5.3 ENTITY ALIGNMENT RESULTS

Table 1 depicts the entity alignment results on the JAPE dataset. We observe that: (1) decentRL significantly outperformed all the methods on Hits@1 and MRR, which empirically showed the advantage of decentRL in learning high-quality representations. (2) The scores of Hits@10 of decentRL were slightly below those of AliNet. We argue that decentRL is a purely end-to-end model, which did not incorporate any additional data augmentation used in AliNet (Sun et al., 2020) that may improve the Hits@10 results. Moreover, decentRL is much easier to be optimized, as it does not need to coordinate the hyper-parameters of each part in the pipeline. Also, there is no conflict to combine decentRL and the data augmentation algorithm for further improvement.

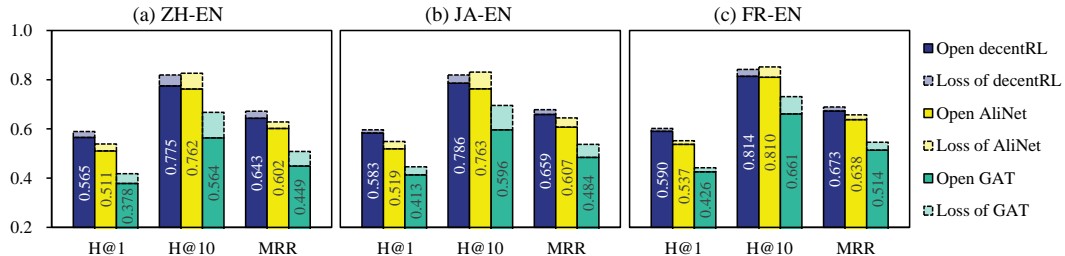

Figure 2: Open entity alignment results on DBP15K. Bars with dotted lines denote the performance drop compared with the corresponding results in the non-open setting. The same to the following.

Table 2: Entity prediction results on FB15K-237.

| Models | TransE | | | DistMult | | |
|---|---|---|---|---|---|---|
| | Hits@10 | MR | MRR | Hits@10 | MR | MRR |
| Raw | 0.465 | 357 | 0.294 | 0.419 | 354 | 0.241 |
| + D-GCN | 0.469 | 351 | 0.299 | 0.497 | 225 | 0.321 |
| + R-GCN | 0.443 | 325 | 0.281 | 0.499 | 230 | 0.324 |
| + W-GCN | 0.444 | 1,520 | 0.267 | 0.504 | 229 | 0.324 |
| + CompGCN | 0.515 | 233 | **0.337** | 0.518 | 200 | 0.338 |
| + decentRL | **0.521** | 159 | 0.334 | **0.541** | 151 | **0.350** |

Raw denotes the original results of the decoders.

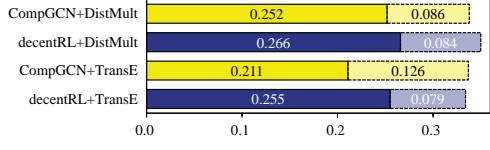

Figure 3: MRR results on open FB15K-237.

We also evaluated the graph-based models in the open-world setting. Specifically, we first split the testing entity set into two subsets, namely known entity set and unknown entity set. Then, for those triples in the training set (in the non-open-world setting, all triples are used in training) containing unknown entities, we moved them to the testing triple set, which were only available during the testing phase. We followed a classical dataset splitting used in the entity alignment task, with 20% of entities in the original testing set are sampled as open entities. Table 6 in Appendix C.1 compares the datasets before and after re-splitting.

The experimental results are shown in Figure 2. We can find that decentRL outperformed GAT and AliNet (the second-best model) on all metrics. Although its performance slightly dropped compared with that in the close setting, the results of others (especially GAT, which only uses self representation as "query") suffered more under this open-world setting. Overall, decentRL is capable of achieving state-of-the-art performance on both open and conventional entity alignment tasks.

## 5.4 ENTITY PREDICTION RESULTS

We also evaluated decentRL on the entity prediction task, in comparison with different GNN-based models: D-GCN (Marcheggiani & Titov, 2017), R-GCN (Schlichtkrull et al., 2018), W-GCN (Shang et al., 2019), and CompGCN (Vashishth et al., 2020). The results on FB15K-237 are shown in Table 2, from which we observe that: (1) decentRL significantly outperformed all the other models for many metrics, especially MR (mean rank). This demonstrates that decentRL can learn better representations for both popular entities (valued by MRR metric) and long-tail entities (valued by MR metric) (2) decentRL boosted DistMult to achieve almost state-of-the-art performance on FB15K-237. The simpler model TransE, also gained great improvement on all metrics. The reason may be that DAN discovered a better aggregation weights and our auto-distiller continuously refined the output representations.

The corresponding results on open entity prediction are shown in Figure 3. We added a state-of-the-art yet more complicated GNN-based model CompGCN + ConvE in the comparison, from which we observe that: decentRL + DistMult outperformed all the other models under this open-world setting, which verified its effectiveness in inductive learning with only structure data. decentRL + TransE achieved the second-best performance, followed by CompGCN + ConvE. Overall, decentRL provided the decoders with better representations and supported them to achieve competitive and even better performance over the cutting-edge models.

Table 3: Entity prediction results on FB15K and WN18.

| Methods | FB15K | | | | | WN18 | | | | |
|---|---|---|---|---|---|---|---|---|---|---|
| | Hits@1 | Hits@3 | Hits@10 | MRR | MR | Hits@1 | Hits@3 | Hits@10 | MRR | MR |
| TransE | 0.297 | 0.578 | 0.749 | 0.463 | - | 0.113 | 0.888 | 0.943 | 0.495 | - |
| DistMult | 0.546 | 0.733 | 0.824 | 0.654 | 97 | 0.728 | 0.914 | 0.936 | 0.822 | 902 |
| ComplEx | 0.599 | 0.759 | 0.840 | 0.692 | - | 0.599 | 0.759 | 0.840 | 0.692 | - |
| ConvE | 0.558 | 0.723 | 0.831 | 0.657 | 51 | 0.935 | 0.946 | 0.956 | 0.943 | 374 |
| RotatE | 0.746 | 0.830 | 0.884 | 0.797 | 40 | 0.944 | 0.952 | 0.959 | 0.949 | 309 |
| RSN | 0.722 | - | 0.873 | 0.78 | - | 0.922 | - | 0.953 | 0.940 | - |
| TuckER | 0.741 | 0.833 | 0.892 | 0.795 | - | 0.949 | 0.955 | 0.958 | 0.953 | - |
| decentRL + TransE | 0.633 | 0.771 | 0.856 | 0.715 | 40 | 0.736 | 0.904 | 0.954 | 0.824 | 255 |
| decentRL + DistMult | 0.664 | 0.793 | 0.872 | 0.740 | **32** | 0.944 | 0.951 | 0.958 | 0.949 | 259 |
| decentRL + ComplEx | 0.745 | **0.847** | **0.901** | **0.804** | 33 | 0.945 | 0.952 | 0.958 | 0.949 | **251** |

Table 4: Entity prediction results on FB15K-237 and WN18RR.

| Methods | FB15K-237 | | | | | WN18RR | | | | |
|---|---|---|---|---|---|---|---|---|---|---|
| | Hits@1 | Hits@3 | Hits@10 | MRR | MR | Hits@1 | Hits@3 | Hits@10 | MRR | MR |
| TransE | - | - | 0.465 | 0.294 | 357 | - | - | 0.501 | 0.226 | 3,384 |
| DistMult | 0.155 | 0.263 | 0.419 | 0.241 | 254 | 0.39 | 0.44 | 0.49 | 0.43 | 5,110 |
| ComplEx | 0.158 | 0.275 | 0.428 | 0.247 | 339 | 0.41 | 0.46 | 0.51 | 0.44 | 5,261 |
| ConvE | 0.237 | 0.356 | 0.501 | 0.325 | 244 | 0.400 | 0.440 | 0.520 | 0.430 | 4,187 |
| RotatE | 0.241 | 0.375 | 0.533 | 0.338 | 177 | 0.428 | 0.492 | 0.571 | 0.476 | 3,340 |
| RSN | 0.202 | - | 0.453 | 0.280 | - | - | - | - | - | - |
| TuckER | 0.266 | 0.394 | **0.544** | 0.358 | - | 0.443 | 0.482 | 0.526 | 0.470 | - |
| CompGCN + ConvE | 0.264 | 0.390 | 0.535 | 0.355 | 197 | 0.443 | 0.494 | 0.546 | 0.479 | 3,533 |
| CompGCN + TransE[†] | 0.242 | 0.367 | 0.510 | 0.332 | 214 | - | - | - | - | - |
| CompGCN + DistMult[†] | 0.249 | 0.368 | 0.515 | 0.337 | 199 | - | - | - | - | - |
| CompGCN + ConvE[†] | 0.262 | 0.385 | 0.532 | 0.352 | 215 | - | - | - | - | - |
| decentRL + TransE | 0.241 | 0.362 | 0.521 | 0.334 | 159 | 0.290 | 0.420 | 0.505 | 0.369 | 4,710 |
| decentRL + DistMult | 0.257 | 0.385 | 0.541 | 0.350 | **151** | 0.433 | 0.481 | 0.542 | 0.470 | 4,613 |
| decentRL + ComplEx | 0.261 | 0.388 | **0.544** | 0.354 | 172 | 0.422 | 0.466 | 0.533 | 0.458 | 3,744 |

"†" denotes methods executed by the source code with the provided best parameter settings.

Table 5: Ablation study of entity alignment on DBP15K (average of 5 runs).

| Models | ZH-EN | | | JA-EN | | | FR-EN | | |
|---|---|---|---|---|---|---|---|---|---|
| | Hits@1 | Hits@10 | MRR | Hits@1 | Hits@10 | MRR | Hits@1 | Hits@10 | MRR |
| decentRL + auto-distiller | **0.589** | **0.819** | **0.672** | **0.596** | **0.819** | **0.678** | **0.602** | **0.842** | **0.689** |
| decentRL + infoNCE | 0.579 | 0.816 | 0.665 | 0.591 | 0.816 | 0.673 | 0.593 | 0.834 | 0.682 |
| decentRL + L2 | 0.571 | 0.802 | 0.655 | 0.589 | 0.807 | 0.669 | 0.591 | 0.831 | 0.679 |
| centRL + auto-distiller | 0.579 | 0.812 | 0.663 | 0.589 | 0.812 | 0.671 | 0.593 | 0.836 | 0.681 |
| centRL | 0.544 | 0.791 | 0.632 | 0.561 | 0.799 | 0.646 | 0.560 | 0.820 | 0.654 |

The detailed results on entity prediction are shown in Tables 3 and 4, respectively. For the conventional benchmarks FB15K and WN18 that have been widely used for many years, our decentRL with only simple decoders achieved competitive even better performance compared with those state-of-the-art models. Furthermore, decentRL greatly improved the best results on MR, as it can more efficiently aggregate neighbor information to learn high-quality representations for those "challenging" entities.

On the other hand, we find that the performance of decentRL on FB15K-237 and WN18RR is not as promising as that in Table 3, although it still achieved the best Hits@10 and MR on FB15K-237. We argue that this may be caused by the insufficient predictive ability of simple decoders (Guo et al., 2019). However, we currently do not plan to adapt decentRL to some complex decoders like ConvE, as such complicated architecture can largely increase the time and space complexity. For example, CompGCN with ConvE needs to be trained at least two days on a small dataset like FB15K-237.

Overall, the performance of some simple linear KG representation learning models (i.e., TransE, DistMult, and ComplEx) received great benefits from decentRL, and even outperformed some cutting-edge models.

Figure 4: Hits@1 results of each layer and the concatenation. The results of AliNet are from (Sun et al., 2020). It has no L3 and L4 scores as its best performance was achieved by a two-layer model.

## 5.5 Comparison with Alternative Models

To exemplify the effectiveness of each module in decentRL, we derived a series of alternative models from decentRL and report the experimental results on entity alignment in Table 5. "centRL" denotes the model that used DAN but self-loop was added to the adjacency matrix.

From the results, we observe that: all the models in the table achieved state-of-the-art performance on Hits@1 metric, as DAN which leverages all neighbors as queries can better summarize the neighbor representations of entities.

On the other hand, we also find that decentRL + auto-distiller outperformed all the other alternatives. The centralized model centRL + auto-distiller had a performance drop compared with the decentralized one. The main reason is that entities themselves in centRL also participated in their own aggregations, which disturbed the roles of the original representations. Please see Appendix C.2 for the corresponding results under the open entity alignment setting.

## 5.6 Comparison of the Output Embeddings of Each Layer

We also compared the performance of each layer in decentRL and AliNet. As shown in Figure 4, decentRL consistently outperformed AliNet on each layer except the input layer. As mentioned before, decentRL does not take the original representation of an entity as input, but this representation can still gain knowledge in participating in the aggregations of its neighbors and then teach the student (i.e., the corresponding output representation). The performance of the input layer was not as good as that in AliNet, because the latent information in this layer may not be aligned in each dimension.

On the other hand, we also observe that concatenating the representations of each layer in decentRL also improved the performance, with a maximum increase of 0.025 (0.023 for AliNet). Furthermore, decentRL can gain more benefits from increasing the layer number, while the performance of AliNet starts to drop when the layer number is larger than 2 (Sun et al., 2020).

## 6 Conclusion

In this paper we proposed decentralized KG representation learning, which explores a new and straightforward way to learn representations in open-world, with only structure information. The corresponding end-to-end framework achieved very competitive performance on both entity alignment and entity prediction tasks.

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

# A  DECENTRALIZED ATTENTION NETWORKS

## A.1  ARCHITECTURE

We illustrate a four-layer decentralized attention network (DAN) in Figure 5. Following the existing works AliNet (Sun et al., 2020), CompGCN (Vashishth et al., 2020) etc. (Ye et al., 2019; Wu et al., 2019), we also combine the relation embeddings in the aggregation. The original entity embeddings (i.e., $\mathbf{g}^{-1}$), relation embeddings and weight matrices are randomly initialized before training. At step 0, we initialize $\mathbf{g}^0$ with the original entity embeddings by mean aggregator. At step 1, $\mathbf{g}^{-1}$ and $\mathbf{g}^0$ are fed into DAN. Then, we combine the hidden representations with relation embeddings (steps 2, 3). Finally, we obtain the output of the first layer $\mathbf{g}^1$ (step 4). Repeating steps 1–4, we can sequentially obtain the output of the last layers.

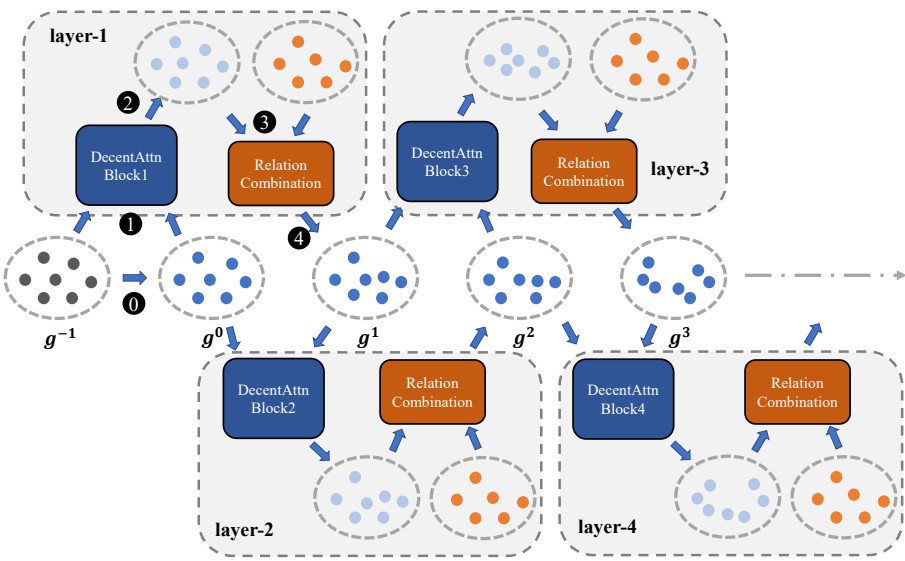

Figure 5: Overview of a four-layer decentralized attention network (DAN). Best viewed in color. Grey, blue, light blue and orange nodes denote original entity embeddings, decentralized entity embeddings, hidden entity embeddings and relation embeddings, respectively. Taking the first layer as example, as DAN requires the output embeddings of two previous layers as input, we randomly initialize the original embeddings as $\mathbf{g}^{-1}$ (identical to $\mathbf{d}^{-1}$), and use an appropriate aggregator to generate the initial decentralized embeddings $\mathbf{g}^0$ (identical to $\mathbf{d}^0$).

## A.2  IMPLEMENTATION DETAILS

**Infrastructure.** Following GAT (Velickovic et al., 2018), we adopt dropout (Srivastava et al., 2014) and layer normalization (Ba et al., 2016) for each module in DAN. To fairly compare decentRL with other models, we do not leverage the multi-head attention mechanism (Vaswani et al., 2017; Velickovic et al., 2018) which has not been used in other GNN-based models, although it can be easily integrated.

Furthermore, we consider residual connections (He et al., 2016) between different layers of DAN to avoid over-smoothing, in which we not only consider the output of the previous layer as "residual", but also involve the output of the mean aggregator (i.e., $\mathbf{g}_i^0$). This can be written as follows:

$$\mathbf{g}_i^{k'} := \mathbf{g}_i^0 + \mathbf{g}_i^{k-1} + \mathbf{g}_i^k. \tag{10}$$

For simplicity, in the rest of the paper, we still use $\mathbf{g}_i^k$ to denote the output at layer $k$.

**Adaption to different tasks.** For different KG representation learning tasks, we also consider different adaptation strategies to achieve better performance.

For the entity alignment task, we follow the existing GNN-based models (Sun et al., 2020; Wang et al., 2018) to concatenate the output representation of each layer as the final output. We formalize it as follows:

$$\mathbf{g}_i = [\mathbf{g}_i^1 \parallel \ldots \parallel \mathbf{g}_i^K]. \tag{11}$$

On the other hand, the entity prediction task prefers the prediction ability rather than the learned representations (Guo et al., 2019). We only use the output of the last layer as the final output representation, which allows us to choose larger batch size or hidden size to obtain better performance. We write it as follows:

$$\mathbf{g}_i = \mathbf{g}_i^K. \tag{12}$$

To enhance the predictive ability of decoders, here we only regard the mutual information-based loss as a kind of regularization (which is similar to MINE (Belghazi et al., 2018)), and thus we re-scale the loss weight to 0.001.

## B  Automatic Knowledge Distillation

### B.1  Insight

The existing works usually choose to jointly optimize two input variables in the density function $f$, in which these two variables can be regarded as two different outputs of two models. For example, InfoNCE uses an encoder to obtain the latent representations and another model to summarize those representations to one context vector. This is similar to DeepInfoMax and DeepGraphInfo, which also leverage two models to obtain local features and summarize global features, respectively.

However, in our case, the mutual information that we want to maximize is between the input and output of the same model, where the input vectors are just randomly initialized raw embeddings. We argue that jointly optimizing the original input $\mathbf{e}_i$ with the output $\mathbf{g}_i$ in Equation (7) may drive $\mathbf{e}_i$ to completely match $\mathbf{g}_i$.

To resolve this problem, we only use the copy of $\mathbf{e}_i$ when estimating the mutual information density between $\mathbf{e}_i$ and $\mathbf{g}_i$. In other words, we do not update the gradient of $\mathbf{e}_i$ in Equation (7), leading to a natural knowledge distillation architecture.

Specifically, we separately optimize $\mathbf{e}_i$ and $\mathbf{g}_i$ in different training examples or batches. The first step is corresponding to the former part of Equation (8):

$$\underset{\mathbf{g}_i, f}{\operatorname{argmax}} \underset{X_i}{\mathbb{E}} \log \Big( \frac{f(\mathbf{g}_i, \hat{\mathbf{e}}_i)}{\sum_{e_j \in X_i} f(\mathbf{g}_i, \hat{\mathbf{e}}_j)} \Big). \tag{13}$$

Here, $\mathbf{e}_i$ is served as a "pre-trained" teacher model to teach a "student". Hence, the learnable parameters are $\mathbf{g}_i$ and $f$.

As aforementioned, $\mathbf{e}_i$ needs to participate in learning the representations of its neighbors, during which it can gain knowledge to teach its student $\mathbf{g}_i$. This step is achieved by the latter part in Equation (8):

$$\underset{\mathbf{e}_i}{\operatorname{argmax}} \sum_{e_j \in N_i} \underset{X_j}{\mathbb{E}} \log \Big( \frac{f(\mathbf{g}_j, \hat{\mathbf{e}}_j)}{\sum_{e_k \in X_j} f(\mathbf{g}_j, \hat{\mathbf{e}}_k)} \Big), \tag{14}$$

where our aim is to find the optimal $\mathbf{e}_i$ to maximize the mutual information between the original and output representations of its neighbors.

### B.2  The Lower-bound of Mutual Information

We do not really need to explicitly separate the training procedure into the two steps described in Appendix B.1, which is widely used in adversarial learning. Instead, this knowledge distillation mechanism can be automatically achieved during different mini-batches.

Specifically, if we expand Equation (13) a little bit, then we obtain:

$$(\mathbf{N}_i, \Theta, f) = \operatorname*{argmax}_{\mathbf{N}_i, \Theta, f} \ \mathbb{E}_{X_i} \log \Big( \frac{f(G(\mathbf{N}_i), \hat{\mathbf{e}}_i)}{\sum_{e_j \in X_i} f(G(\mathbf{N}_i), \hat{\mathbf{e}}_j)} \Big), \tag{15}$$

where $\mathbf{N}_i = \{\mathbf{e}_j | e_j \in N_i\}$ is the original neighbor embedding set for $\mathbf{e}_i$ and $\Theta$ denotes the parameters of our decentralized model $G$. As the optimal $\Theta$ for the model depends on the neighbor representation set $\mathbf{N}_i$, and the optimal density function $f$ also relies on the output of the model, it is impossible to search all spaces to find the best parameters. In practice, we choose to optimize a weaker lower-bound on the mutual information $I(\mathbf{g}_i, \hat{\mathbf{e}}_i)$ (Tian et al., 2020). In this case, a relatively optimal neighbor embedding $\mathbf{e}_x^*$ in Equation (15) is:

$$\mathbf{e}_x^* = \operatorname*{argmax}_{\mathbf{e}_x} \ \mathbb{E}_{X_i} \log \Big( \frac{f(G(\mathbf{N}_i), \hat{\mathbf{e}}_i)}{\sum_{e_j \in X_i} f(G(\mathbf{N}_i), \hat{\mathbf{e}}_j)} \Big), \tag{16}$$

and we have:

$$\widehat{I}(\mathbf{g}_i, \hat{\mathbf{e}}_i | \mathbf{e}_x^*) = \mathbb{E}_{X_i} \log \Big( \frac{f(G(\{\mathbf{e}_1, \dots, \mathbf{e}_x^*, \dots, \mathbf{e}_{|N_i|}\}, \hat{\mathbf{e}}_i)}{\sum_{e_j \in X_i} f(G(\{\mathbf{e}_1, \dots, \mathbf{e}_x^*, \dots, \mathbf{e}_{|N_i|}\}, \hat{\mathbf{e}}_j)} \Big) \tag{17}$$

$$\leq \mathbb{E}_{X_i} \log \Big( \frac{f^*(G^*(\mathbf{N}_i^*), \hat{\mathbf{e}}_i)}{\sum_{e_j \in X_i} f^*(G^*(\mathbf{N}_i^*), \hat{\mathbf{e}}_j)} \Big) = \widehat{I}(\mathbf{g}_i^*, \hat{\mathbf{e}}_i) \tag{18}$$

$$\leq \widehat{I}(\mathbf{g}_i^*, \hat{\mathbf{e}}_i) + \log(|X_i|) \tag{19}$$

$$\leq I(\mathbf{g}_i, \hat{\mathbf{e}}_i), \tag{20}$$

where $^*$ denotes the optimal setting for the corresponding parameters. Equations (18) and (19) are the conclusion of InfoNCE, given that $|X_i|$ is large enough. The above equations suggest that optimizing $\mathbf{e}_x$ can also lower-bound the mutual information without the requirement of other parameters being perfectly assigned.

Consider that the entity $e_x$ may have more than one neighbor, we can optimize those cases together:

$$\mathbf{e}_x^* = \operatorname*{argmax}_{\mathbf{e}_x} \sum_{e_j \in N_x} \mathbb{E}_{X_j} \log \Big( \frac{f(G(\mathbf{N}_j), \hat{\mathbf{e}}_j)}{\sum_{e_k \in X_j} f(G(\mathbf{N}_j), \hat{\mathbf{e}}_k)} \Big) \tag{21}$$

Evidently, the above equation is identical to Equation (14), which means that optimizing Equation (15) can subsequently contribute to optimizing the original neighbor representations.

Therefore, the proposed architecture can automatically distill knowledge, in different mini-batches, from the original representations into the output representations.

## C  FURTHER ANALYSIS

### C.1  DATASET DETAILS

The detailed statistics of the entity alignment datasets are shown in Table 6. Although we only set 20% of entities in testing set as open entities, there are actually more than 20% of triples that were removed from the training set.

For the details of datasets used in entity prediction, we suggest readers to refer to (Bordes et al., 2013) and (Dettmers et al., 2018).

### C.2  ABLATION STUDY ON OPEN ENTITY ALIGNMENT

We also conducted an ablation study on the open entity alignment task, as shown in Table 7. The experimental results, in principle, are consistent to those on conventional entity alignment. The proposed architecture (decentRL + auto-distiller) still outperformed other alternatives. By contrast, the performance of the centralized model with auto-distiller dropped significantly, in comparison with that it almost has identical performance with decentRL + infoNCE in Table 5. Another worth-noting point is that the gap on Hits@10 narrowed in the open entity alignment task, which may be because the training data were shrunk considerably due to removing the corresponding triples referred to unseen entities.

Table 6: Statistics of entity alignment datasets.

| Datasets | Original | | | | | Open | |
|---|---|---|---|---|---|---|---|
| | #Entities | #Relations | #Triples | #Train entity pairs | #Test entity pairs | #Train triples | #Test triples |
| ZH-EN | 19,388 | 1,701 | 70,414 | 4,500 | 10,500 | 53,428 | 16,986 |
| | 19,572 | 1,323 | 95,142 | 4,500 | 10,500 | 72,261 | 22,881 |
| JA-EN | 19,814 | 1,299 | 77,214 | 4,500 | 10,500 | 57,585 | 19,629 |
| | 19,780 | 1,153 | 93,484 | 4,500 | 10,500 | 69,479 | 24,005 |
| FR-EN | 19,661 | 903 | 105,998 | 4,500 | 10,500 | 79,266 | 26732 |
| | 19,993 | 1,208 | 115,722 | 4,500 | 10,500 | 87,030 | 28692 |

Table 7: Ablation study on open entity alignment. Average of 5 runs.

| Methods | ZH-EN | | | JA-EN | | | FR-EN | | |
|---|---|---|---|---|---|---|---|---|---|
| | Hits@1 | Hits@10 | MRR | Hits@1 | Hits@10 | MRR | Hits@1 | Hits@10 | MRR |
| decentRL + auto-distiller | **0.565** | **0.775** | **0.643** | **0.583** | **0.786** | **0.659** | **0.590** | **0.814** | **0.673** |
| decentRL + infoNCE | 0.557 | **0.775** | 0.637 | 0.574 | 0.785 | 0.652 | 0.583 | 0.811 | 0.666 |
| decentRL + L2 | 0.552 | 0.770 | 0.632 | 0.574 | 0.782 | 0.650 | 0.581 | 0.806 | 0.664 |
| centRL + auto-distiller | 0.551 | 0.765 | 0.629 | 0.573 | 0.776 | 0.648 | 0.578 | 0.806 | 0.662 |
| centRL | 0.529 | 0.764 | 0.614 | 0.554 | 0.775 | 0.634 | 0.560 | 0.799 | 0.647 |

Table 8: Performance of decentRL with different dimensions. Average of 5 runs.

| Hidden size | ZH-EN | | | JA-EN | | | FR-EN | | |
|---|---|---|---|---|---|---|---|---|---|
| | Hits@1 | Hits@10 | MRR | Hits@1 | Hits@10 | MRR | Hits@1 | Hits@10 | MRR |
| 64 | 0.429 | 0.644 | 0.508 | 0.474 | 0.673 | 0.547 | 0.468 | 0.704 | 0.554 |
| 128 | 0.511 | 0.726 | 0.590 | 0.541 | 0.745 | 0.617 | 0.535 | 0.773 | 0.623 |
| 256 | 0.560 | 0.785 | 0.643 | 0.578 | 0.791 | 0.657 | 0.578 | 0.817 | 0.665 |
| 512 | **0.589** | **0.819** | **0.672** | **0.596** | **0.819** | **0.678** | **0.602** | **0.842** | **0.689** |

## C.3 IMPACT OF DIMENSIONS

We also evaluated decentRL under different settings of dimensions. The results are shown in Table 8. With the increase of the input dimensions (i.e., embedding size), the performance of decentRL improved quickly, with dimension $= 128$ achieving comparable performance with the state-of-the-art methods (e.g., AliNet with 300 dimension) and outperforming them at dimension $= 256$. Furthermore, decentRL can continually gain benefit from larger hidden sizes. Even when the dimension was set to $512$, the improvement was still significant.

