# OpenReview forum: "Decentralized Knowledge Graph Representation Learning"
_ICLR.cc/2021/Conference — Reject_

### Official Review · AnonReviewer2 · 2020-10-14
**Promising results, but paper very hard to understand and follow**

**Rating:** 4
**Confidence:** 4

**Review:**

This work proposes a GNN-based model for learning KG embeddings purely from the embeddings of its neighbours, which would enable learning entity embeddings for previously unseen entities. The model is based on a modified version of a graph attention network (GAT) which only considers the embeddings of neighbouring nodes.

Despite the advantage of decentRL over existing approachs of its applicability on previously unseen entities and the results on entity alignment and entity/link prediction showing a lot of promise, I've found the paper quite hard to follow due to factual inaccuracies and English style and grammar issues, which is why I believe it is not ready for publication in its current form. I encourage the authors to revise the paper and resubmit to the next big ML/graphs conference.


Detailed comments and questions below:

Sec. 1\
The statement "then TransE and many KG embedding models (Wang et al., 2014; Dettmers et al., 2018; Nguyen et al., 2018; Kazemi & Poole, 2018; Sun et al., 2019), learn representations in a Skip-gram (Mikolov et al., 2013a) manner" is factually incorrect, since the data and learning objective for learning word embeddings differ greatly from those for learning KG embeddings. Analogical properties of word embeddings are an implicit bi-product of the skip-gram training objective, whereas in e.g. TransE this property is explicitly imposed on relation representations through its score function. The score function of other KG embedding models mentioned (e.g. ConvE, Dettmers et al. 2018 or SimplE, Kazemi & Poole 2018) is of a different type and does not even impose the analogical property of relations.

Sec. 3.1\
"Intuitively, if e_i is the embedding of an unseen entity, it is rarely useful in computing the attention scores (as it is just a randomly initialized vector). Thus, purely relying on its neighbors may be a good choice." - Aren't the neigbours randomly initialised as well?

Sec. 3.2\
After being introduced, n_i is not used anywhere. What is its role in Eq. 3?

Sec. 5.4
- As one of two main results sections, entity prediction results which include standard link prediction models (Tables 7 and 8) should be moved from the Appendix into the main body.
- Given that TransE and DistMult haven't been state-of-the-art for quite a while, it would be interesting to see how decentRL performs with recent state-of-the-art link prediction models, such as TuckER or RotatE.

Other comments:

Writing quality should be improved, as the paper is hard to follow. Please have your submission proof-read for English style and grammar issues.

===============================================================================================================\
After rebuttal:\
I have read the authors' response, but since the actual body of the paper has not changed much from the original submission, I stand by my original rating.

---

> ### Author Response · Authors · 2020-11-17
> **Response to Reviewer #2**
>
> Thanks for your suggestion, we address the main concerns below:
>
> Sec. 1: The statement "then TransE and many KG embedding models ..., in a Skip-gram (Mikolov et al., 2013a) manner" is factually incorrect, since the data and learning objective for learning word embeddings differ greatly from those for learning KG embeddings.
>
> \- When we talk about ''a skip-gram manner'', we have no intention of highlighting the similarity of objectives or algorithms. We actually want to point out that they evaluate the models in the same form (i.e., one-to-many, or many-to-one). We have reclarified this in the revision.
>
> Sec. 3.1: "Intuitively, if e_i is the embedding of an unseen entity, it is rarely useful in computing the attention scores (as it is just a randomly initialized vector). Thus, purely relying on its neighbors may be a good choice." - Aren't the neigbours randomly initialised as well?
>
> \- In our assumption, there only exist a relatively small proportion of unseen entities. In other words, we have most neighbor embeddings of an unseen entity after training.
>
> Sec. 3.2: After being introduced, $\textbf{n}_i$ is not used anywhere. What is its role in Eq. 3?
>
> \- We used here to better illustrate how a common solution could be. The output $\textbf{d}_i^{k-1}$ of layer k-1 is actually the neighbor context $\textbf{n}_i$ for layer k-2.
>
> Sec. 5.4:
>
> 1\. As one of two main results sections, entity prediction results which include standard link prediction models (Tables 7 and 8) should be moved from the Appendix into the main body.
>
> \- Good idea, we have moved them into the main text.
>
> 2\. Given that TransE and DistMult haven't been state-of-the-art for quite a while, it would be interesting to see how decentRL performs with recent state-of-the-art link prediction models, such as TuckER or RotatE.
>
> \- There are two reasons for choosing TransE and DistMult as decoder: 1) the current state-of-the-art methods usually take it as decoder, which means that we can fairly compare the results; 2) they are more efficient, especially considering that we already leverage a complicated GCN model as encoder.
>
> Other comments: Writing quality should be improved, as the paper is hard to follow. Please have your submission proof-read for English style and grammar issues.
>
> \- We will thoroughly check and fix grammatical errors in the final submission.

---

### Official Review · AnonReviewer4 · 2020-10-22
**Better justification of the motivation and approach**

**Rating:** 5
**Confidence:** 4

**Review:**

This paper presents a method for knowledge graph embedding based on graph attention networks (GAT). The key idea is to avoid using the information for a node (i.e., its representation vectors) when computing the attention weights for the neighbors of the node. The paper argues that this approach can better generalize to unseen nodes where no pre-defined features/information is available. As such, the paper does not include the representations for a node $e$ from prior layers in the aggregations to compute $e$'s representations in the next layers, leveraging the representation vectors of the nodes from prior layers to obtain attention weights for the current layer. The paper also proposes to a self-learning method to learn the parameters by optimizing the mutual information of the final and initial embedding vectors for the nodes. A distillation approach is also employed to use the initial embedding vectors as the teachers and the final embedding vectors as the students. The proposed method is applied to two downstream tasks, i.e., entity alignment and entity prediction, leading to competitive performance with many prior works (the learned node embeddings still need to be aligned using task-specific losses). Some experiments on unseen entities and ablation studies are also conducted to demonstrate the benefits of the proposed method.

Overall, this paper is well written. It introduces an extension of GAT to address the unseen entity issue and the experiments seem to demonstrate its benefit. However, the motivation and approach of the paper should be better justified to make it more convincing. I have several comments/questions as follow:

1. The technical novelty of the paper seems incremental as the DAN mechanism is a simple extension of GAT while mutual information maximization and the distillation are already applied in prior work.

2. The paper seems to assume that for unseen entities, although pre-trained embeddings are not available, unseen entities are still connected to some seen entities so unseen entities' embeddings can still be obtained via the averages of the embeddings of the seen neighbors. As such, how do we handle two unseen entities that are neighbors of each other? More importantly, as the current method does not use any information specific to the nodes, e.g., node content (so only the connections of nodes are employed), can we just include the unseen entities in the graph and retrain the whole model? This is certainly more expensive, but as the paper is mainly considering downstream task performance, this might be a method to address unseen entities in this work. In general, as node embeddings are initialized randomly without considering node content in this work, the proposed method for unseen entities does not seem significant and convincing to me.

3. Relatedly, as the node embeddings \textbf{e}_i are randomly initialized, what kind of knowledge can \textbf{g}_i expect to learn from considering \textbf{e}_i as a teacher? In general, without the guidance from some specific downstream tasks, it is unclear which information the model would learns when the training finishes. Maybe the auto-distiller should be jointly trained with downstream tasks and the model can be better justified with the idea from information bottleneck? A discussion about the connection of the proposed method and information bottleneck is also helpful.

4. How does the performance changes if we directly use \textbf{e}_i (not its copy) in Equations (7) and (8)? This might help to better justify the model design.

---

> ### Author Response · Authors · 2020-11-17
> **Response to Reviewer #4**
>
> Thanks for your helpful comments, we wish the following reply can erase your concerns:
>
> Q1: The technical novelty of the paper seems incremental as the DAN mechanism is a simple extension of GAT while mutual information maximization and the distillation are already applied in prior work.
>
> \- As the title shows, the key contribution of this paper is that we propose a new approach (not merely an extension of GAT) to learn “decentralized” representations. DAN is one proper implementation of this approach, and we can adapt the “decentralized” idea to many GNN models, such as the general GCNs or state-of-the-art models in KG representation learning area. auto-distiller is the key to enable the model to learn how to summarize the neighbor embeddings for unseen entities. Table 3 and Table 5 also showed its effectiveness compared with infoNCE.
>
> Q2: How do we handle two unseen entities that are neighbors of each other?
>
> \- Ignoring this case is sufficient to achieve state-of-the-art performance, as the attention mechanism exists. However, you can choose to create a special mask matrix to exclude them in calculation.
>
>
> Q3: Relatedly, as the node embeddings $\textbf{e}_i$ are randomly initialized, what kind of knowledge can \textbf{g}_i expect to learn from considering $\textbf{e}_i$ as a teacher? Maybe the auto-distiller should be jointly trained with downstream tasks.
>
> \- Exactly, auto-distiller was jointly trained with the downstream tasks like entity alignment and entity prediction in this paper (please see Sec. 5.2). We also clarify this key attribution in Lemma 1. In short, training $\textbf{g}_i$ will update the embeddings of $e_i$ ’s neighbors. Then, we can distill the knowledge of $\textbf{e}_j$ (one neighbor of $e_i$) into $\textbf{g}_j$.
>
> Q4: How does the performance changes if we directly use $\textbf{e}_i$ (not its copy) in -Equations (7) and (8)? This might help to better justify the model design.
>
> \- It is the original InfoNCE if we do not use the copy. We presented this result in the ablation study section (Table 3).

---

### Official Review · AnonReviewer1 · 2020-10-28
**OK method, but significant clarity issues.**

**Rating:** 4
**Confidence:** 3

**Review:**

=== Summary ===

This paper proposes a "decentralized" method for representation learning in knowledge graphs that doesn't explicitly depend on a learned embedding for the entity node of interest, e_i. Rather, the embedding for e_i is constructed in a distributed fashion (similar in motivation to the distributional hypothesis/skip-gram word embeddings) from its neighbors via a second-order attention mechanism. The main idea is that this is better for "cold start" problems in which unknown entities might have no features, which makes building any representation that explicitly depends on entity-centric features hard.

=== Justification for Score ===

While the method might indeed hold some promise, I find it a bit difficult to get excited about it. Furthermore, in the current version it's presentation and motivation is unclear and hard to follow. It is also not clear from the paper why this method should work better than simpler baselines (see concerns).

=== Strengths ===

- Relevant and timely topic, as leveraging knowledge graphs with initially unknown entities etc is an important problem (especially with constantly growing KGs).

- Some mixed empirical results w.r.t. other compared models, but generally positive.

=== Concerns ===

- It's unclear to me why a purely "decentralized" representation is desired, especially if the experiments aren't purely on unknown entity settings. A natural baseline to me would seem to be a standard GCN/GAT with "entity dropout", i.e., during training time you introduce entities for which some redundancy must be gained by its neighborhood. Another approach would be to use a framework reminiscent of label propagation to deal only with imputing missing features at test time. Fully dropping all entity-specific features seems like overkill, and potentially harmful. After all, at test time, a majority of the KG will be known.

- In general, in addition to the comment above, though I am not intimately familiar with their details, it appears that none of the considered baseline methods (AliNet, RSN, etc) are specifically designed to accommodate missing entities. As this is a major claimed contribution, it would be useful to either clarify this, or explain how it is handled differently in these networks.

- I'm a bit perplexed by the consistently underperforming H@10 results relative to AliNet---and I'm not sure I find the justification in the paper (data augmentation) that convincing (can you clarify why this would explain worse performance @10, but not @1 or MRR?).

- Overall, though there might be something here (I am willing to be convinced otherwise...), the paper fails to convince me of its significance at this stage. I feel that it would benefit greatly from an overall more compelling re-write.

=== Minor Comments ===

- In terms of readability, Table 2 is a bit too small.

- Lemmas 1 and 2 don't add much to the paper in my opinion---I would recommend moving them fully to the appendix.

=== Response After Rebuttal ===

I thank the authors for their responses to my comments. After reading the response as well as the other reviews, I still stand by my original rating. I still find the motivation and empirical results non-compelling, given the current version of the paper.

---

> ### Author Response · Authors · 2020-11-17
> **Response to Reviewer #1**
>
> Thanks for your feedback. We address your concerns as in the following:
>
> Q1: It's unclear to me why a purely "decentralized" representation is desired, especially if the experiments aren't purely on unknown entity settings.
>
> \- As we illustrated in Introduction and Sec. 3.3, decentralized attention mechanism has three advantages: 1) For unseen entities, it does not use the randomly initialized vectors as query. The attention scores are more reliable. 2) For entities with a small number of neighbors, whose embeddings are less informative, using the neighbor context as query is more robust. 3) It does not increase the complexity. Our method is towards KG representation learning, rather than prediction or classification. In other words, we aim to learn low-dimensional vectors for entities. It is weird for a method to embed only unknown entities.
>
> Q2: In general, in addition to the comment above, though I am not intimately familiar with their details, it appears that none of the considered baseline methods (AliNet, RSN, etc) are specifically designed to accommodate missing entities.
>
> \- Indeed, open world KG embedding is still a challengeable yet uncharted task. Some methods explore specific tasks like relation prediction or triple classification under this setting, but none supports embedding techniques. decentRL is the first method that achieves state-of-the-art performance yet enables representation learning for unknown entities.
>
> Q3: I'm a bit perplexed by the consistently underperforming H@10 results relative to AliNet.
>
> \- Specifically, AliNet leverages a similar bootstrapping algorithm used in BootEA, which will regularly add new seed alignments to the training set. Those new seeds may not be all correct, while can improve the performance on Hits@10 (the relatively inaccurate metric).
>
> Q4: Minor Comments
>
> 1\. Table 2 is a bit too small.
>
> \- Corrected.
>
> 2\. Lemmas 1 and 2 don't add much to the paper in my opinion.
>
> \- They state that decentRL can be optimized in a correct and efficient way. We believe that they are certainly important.

---

### Official Review · AnonReviewer3 · 2020-10-29
**A nice extension of a graph embedding approach, but the review of related work seems partial**

**Rating:** 5
**Confidence:** 2

**Review:**

The paper presents "a decentralized KG representation learning approach", named decentRL, which encodes each entity from and only from the embeddings of its neighbors. This approach can therefore account for new entities that have no known features to initialize its embeddings, but do have known links to other entities in the graph.

The main contributions:
- The paper outlines decentralized attention network (DAN), an adaptation of graph attention network (GAT). GAT considers the direct neighbors of an entity in generating its embedding, and computes attention scores based on similarity between each neighbor and the focus entity. Assuming that no meaningful embedding may be available for new entities, DAT represent each entity via the embeddings of its direct 1-hop neighbors. It then considers its 2-hop entity neighbors for computing attention-based embeddings.
- The paper further adapts the optimization process - alternately optimizing the representation of the target entity and its neighbors.

Experimental results are presented on the task of entity alignment across Knowledge Graphs (KGs) and entity prediction (KG completion), using two datasets per task. Competitive performance is obtained in the general case, with improvements when new entities are considered.

Pros:
- The approach is sensible, intuitive, and yields good results in practice.

Cons:
- My main concern is the review of related work, which seems partial; e.g., the discussion is focused on GAT, but the best competing method in the results section is AliNet, which is nowhere described. Also, there exist other methods like DeepWalk which also rely solely on structure information.
- The paper does not read easily
- It is not clear if the code and data will to be released. At least, the dataset with train-test split should be released for comparison purposes.

Comments and typos:
- can you motivate the open-world scenario, where only structure information is available for new entities, and it has no known features?
- Table1: write Hits@1 instead of H@1
- robuster --> more robust
- Oppositely --> In contrast?

---

> ### Author Response · Authors · 2020-11-17
> **Response to Reviewer #3**
>
> Thanks for your detailed comments. We wish our reply can address your concerns:
>
> Q1: the review of related work, which seems partial; e.g., the discussion is focused on GAT, but the best competing method in the results section is AliNet, which is nowhere described. Also, there exist other methods like DeepWalk which also rely solely on structure information.
>
> \- As discussed in Sec. 1 and Sec. 2, many state-of-the-art methods in KG representation learning area are based on GAT. They have better performance than the vanilla GAT, for example, AliNet and CompGCN. On the other hand, we should notice that deepwalk is actually a model specific to node classification, while both entity alignment and entity prediction require the model to precisely find the target entities from thousands of candidates. They are different.
>
> Q2: The paper does not read easily.
>
> \- Admittedly, the concept of ``decentralized’’ in this paper is not as straightforward as it looks. Correctly formulating this mechanism is non-trivial.
>
> Q3: It is not clear if the code and data will to be released. At least, the dataset with train-test split should be released for comparison purposes.
>
> \- We uploaded the code and datasets when first submitting the paper.
>
> Q4: Comments and typos:
>
> 1\. can you motivate the open-world scenario, where only structure information is available for new entities, and it has no known features?
>
> \- Learning representations only from structural information has been studied for years. Its motivation, or practical significance, is not evaluated by the richness of realistic scenarios. The methods that concentrate on structure information are the fundamental of those towards applications. For example, a lot of methods leverage entity description or attribute information for KG embedding, most of which have to choose an existing model to embed structural information.
>
> 2\. Table1: write Hits@1 instead of H@1
>
> \- Corrected.
>
> 3\. Typos
>
> \- Corrected.

---

### Decision · Program_Chairs · 2021-01-07
**Final Decision**

**Decision:**

Reject

**Comment:**

This paper brings interesting ideas (decentralized setting, auto-distillation) but it does not meet the very high requirements that a publication at ICLR requires.

Three main reasons for that:

1/ Motivation & justification: Ultimately the paper is advocating for a pure decentralized approach "which encodes each entity from and only from the embeddings of its neighbors" with the main motivation being to represent better on unseen entities at training. This is quite radical and leads to a complex model and training procedure for a benefit and justification that are not very clear. Are there that many unseen entities in general? What would periodically retrain the whole model do? The computational cost associated to DecentRL should be discussed with regards to that. Some implementation details in appendix A.2 seems rather critical and are not motivated.

2/ Missing comparisons and references: as noted by several reviewers, it would be helpful to have comparisons of other methods that are dealing with missing entities. Some much simpler heuristics could be tried for instance (retraining the model, averaging neighbors, etc.). A discussion with DeepWalk, that is really an adaptation of CBOW for KG should also be added.

3/ Clarity could be improved. Thanks to reviewers' comments, the clarity has increased but could still be worked on as noted by several reviewers. For instance, the analogy  with CBOW right in the intro is confusing: in the 2nd paragraph, CBOW is used as a common manner for methods that are limited, but in the 3rd paragraph, CBOW is also used as an intuition for DecentRL. Some content from supplementary material like the description in A.1 would add a lot of clarity if added earlier.

We encourage the authors to use the many comments from the reviewers to improve further the paper.